# From Bytes to Ideas:
# Language Modeling with Autoregressive U-Nets

**Mathurin Videau**[*]
Meta AI

**Badr Youbi Idrissi**[*]
Meta AI

**Alessandro Leite**
INSA Rouen Normandy, LITIS

**Marc Schoenauer**
TAU, Inria

**Olivier Teytaud**
Thales - CortAIx-Labs

**David Lopez-Paz**
Meta AI

## Abstract

Tokenization imposes a fixed granularity on the input text, freezing how a language model operates on data and how far in the future it predicts. Byte Pair Encoding (BPE) and similar schemes split text once, build a static vocabulary, and leave the model stuck with that choice. We relax this rigidity by introducing an autoregressive U-Net that learns to embed its own tokens as it trains. The network reads raw bytes, pools them into words, then pairs of words, then up to 4 words, yielding a multi-scale representation of the sequence. At deeper stages, the model must predict further into the future — anticipating the next few words rather than the next byte — so deeper stages focus on broader semantic patterns while earlier stages handle fine details. When carefully tuning and controlling pretraining compute, shallow hierarchies are on par with strong BPE baselines, and deeper hierarchies exhibit a promising trend. Because tokenization now lives inside the model, the same system can handle character-level tasks and carry knowledge across low-resource languages.

## 1 Introduction

Language models are about uncovering patterns in a sequence so they can guess what comes next. Before any of that happens, we must decide what the pieces of that sequence—the *tokens*—actually are. That choice is usually frozen in advance by a *tokeniser* that chops raw text into discrete units long before training begins. Consider the sentence "The quick brown fox." A *character*-level tokeniser feeds the model the stream {T, h, e, ␣, q, u} and asks it to predict the next letter i. A *word*-level tokeniser, in contrast, hands over {The, quick} and expects the model to guess brown in one shot. Finer cuts lead to larger sequences and shorten the look-ahead window, whereas coarser cuts lead to shorter sequences but make each token rarer and harder to compare and predict. Regardless of granularity, some form of tokenisation is unavoidable: a sequence must exist before any Transformer can run.

**Byte-Pair Encoding** (BPE) followed by a simple embedding table is by far the most popular approach. It works by repeatedly merging the most frequent byte sequences in the training text until a preset vocabulary limit is reached. This procedure leaves practitioners with just two intuitive *dials*. The first dial is the *training corpus*: whichever text one feeds the algorithm—English prose, source code, or a multilingual mix—determines which patterns are merged and therefore what the final tokens look like. The second dial is the *vocabulary size*: raising this limit lets the merge process run for more steps, producing longer tokens and shorter sequences at the cost of a larger embedding table and output softmax.

---

[*]Equal contribution
Code open-sourced at https://github.com/facebookresearch/lingua/tree/main/apps/aunet

39th Conference on Neural Information Processing Systems (NeurIPS 2025).

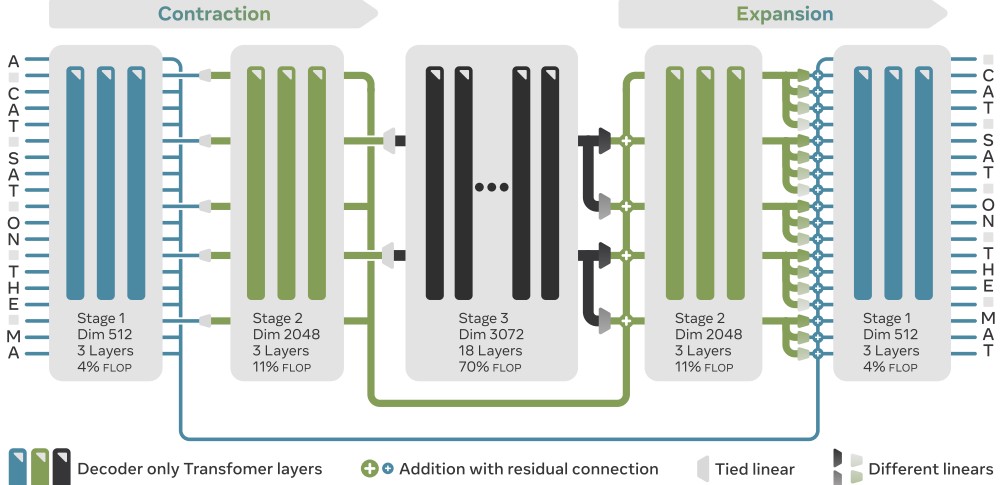

Figure 1: Three-stage Autoregressive U-Net (AU-Net). The model executes from left to right. The contracting path compresses the sequence in two steps: Stage 1 processes raw bytes, Stage 2 keeps only the vector at each word boundary, and Stage 3 keeps one vector per two words. Each contraction and expansion step supports arbitrary pooling and upsampling patterns. After the deepest stage, the expanding path reverses the contracting path by duplicating each coarse vector and applying position-specific linear layers. These are combined with skip connections from the contracting path, gradually restoring sequence length and blending in high-level information. Deeper stages predict further ahead and capture broad semantics, while shallower stages refine local detail.

Most issues with tokenisation stem from the embedding operation rather than the splitting act itself. Each token is typically mapped to an independent vector, meaning the network sees only opaque identifiers and must rediscover, for instance, that *strawberry* and *strawberries* share nine letters. This reliance on isolated embeddings hampers symbol-level reasoning and complicates transfer to dialects or rare languages. Finally, this splitting is most often a preprocessing step, locking in a single level of granularity for all subsequent model layers (see Section 2.2).

To address these limits, our **Autoregressive U-Net** (Section 2.1), or AU-Net ('oh-net', /óʊ nɛt/), learns to embed information directly from raw bytes, and allows for multiple stages of splitting. The purpose of an embedding is to map tokens to vectors. Instead of using a lookup table, we use attention directly to embed the tokens. Self-attention allows vectors at any position to summarize the entire preceding context. This enables a simple pooling mechanism: we select these contextualized vectors at word boundaries (AU-Net-2), then word pairs (AU-Net-3), and up to four-word chunks (AU-Net-4), forming a multi-stage embedding hierarchy. This U-Net like architecture contracts sequences, preserving detail with skip connections, before expanding them. During expansion, vectors representing coarser information are injected back into more fine-grained representations.

Deeper stages operate on compressed representations, allowing them to aggregate information over longer spans of text. While the model remains strictly autoregressive and performs next-byte prediction, the hierarchical structure introduces an inductive bias that encourages the formation of more abstract representations akin to multi-token prediction [1] but achieved without auxiliary losses. This effect allows deeper stages to guide shallower stages at the semantic level, while letting them handle finer details like spelling.

**Contributions** (quantified in Section 3).

**C1.** *Adaptive multi-level hierarchy*. We train up to four end-to-end embedding stages with arbitrary, user-specified split functions, extending prior work that relies either on fixed pooling or shallow hierarchies.

**C2.** *Infinite vocab size*. By operating directly on bytes, our model avoids predefined vocabularies and memory-heavy embedding tables, enabling open-vocabulary modeling without increasing memory footprint. This property is inherent to byte-level approaches, and our results validate its effectiveness at scale.

**C3.** *Strong performance and scaling.* Under identical pre-training budgets, a single level matches strong BPE baselines, and a two or three-level hierarchy shows promising scaling trends. A selection of the results is presented in Table 1

**C4.** *Practical Efficiency.* We maintain comparable GPU throughput in wall-clock time instead of purely theoretical compute gains. Our code is available in Meta Lingua [2].

**C5.** *Stable scaling laws.* We show that moving from token to byte-level training demands new batch size and learning rate formulas to get smooth optimization.

By turning a one-shot, memory-hungry embedding into a learned, multi-scale process, we offer a flexible alternative to the rigid BPE preprocessing followed by a simple embedding table. Table 1 provides a concise overview of results obtained at the 1B scale on 370B tokens, comparing AU-Net-2 (two-stage), AU-Net-3 (three-stage), and AU-Net-4 (four-stage) variants. These results highlight the promising performance in the heavily overtrained regime.

## 2 Method

### 2.1 Autoregressive U-Net

Inspired by U-Net-like architectures [3, 4], we propose an autoregressive hierarchical model for language modelling, illustrated in figure 1. This architecture features a *contracting path*, which compresses the input sequence, and an *expanding path*, which reconstructs it. Both paths are fully *adaptive*: they do not require fixed pooling or upsampling sizes. Pooling and upsampling operations can be designed independently, even

Table 1: 1B equivalent on 370B tokens

| Model | FLOP | Hellaswag | MMLU | GSM8k |
|---|---|---|---|---|
| BPE | 4e21 | 70.2 | 27.0 | 4.4 |
| AU-Net 2 | 3e21 | 69.9 | 28.8 | 3.0 |
| AU-Net 3 | 4e21 | 72.9 | 28.0 | 3.7 |
| AU-Net 4 | 5e21 | **73.7** | **31.7** | **5.3** |

if we choose to make them symmetrical in this paper. The only requirement is a *splitting function*, which specifies the positions in the sequence where pooling should occur. This function is detailed in section 2.2.

Our architecture is *monolithic*: unlike recent approaches [5, 6] that use local models, we apply attention globally at each stage (or within a sliding window), allowing every input to attend to previous inputs. This ensures that words or word groups are not processed in isolation. To preserve fine-grained information that might be lost during contraction, we introduce skip connections between stages, following the approach in Ronneberger et al. [3] and Nawrot et al. [4]. We also increase the hidden dimension at each stage in proportion to its contraction factor, enabling richer representations as the sequence is contracted. To keep computation tractable at the byte-level stage (Stage 1), where sequences are longest, we restrict attention to a window.

#### 2.1.1 Pooling and Upsampling

Since our pooling and upsampling are adaptive, we cannot rely on fixed window sizes. To address this, we explored several pooling and upsampling strategies. In this section, we describe the method used in all experiments reported in the main text. A complete description of the alternatives and ablation results can be found in the appendix C.

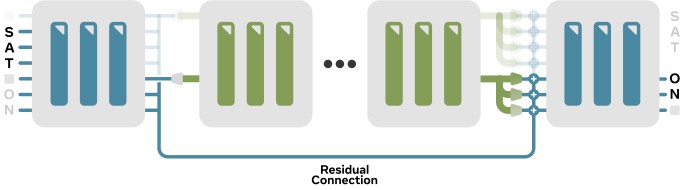

Figure 2: Pooling simply selects the vectors at the positions specified by the splitting function. Upsampling then expands each pooled vector to fill the next segment, applying a separate linear layer for each position. For instance, the pooled vector representing the word 'SAT␣' is used to help predict 'ON␣'. This offset lets deeper stages predict further ahead in the sequence. When using 4 stages, for example, this results in the deepest stage helping for the prediction of the next four words.

**Pooling.** We adopt the simplest pooling strategy: selecting the indices identified by the splitting function and projecting them to the next stage's dimensionality using a linear layer. Since the preceding layers already include attention mechanisms, we rely on these to do the pooling implicitly instead of relying on explicit cross attention as used in Nawrot et al. [4], Pagnoni et al. [5].

Formally, let $X \in \mathbb{R}^{s \times d_{\text{in}}}$ denote the sequence of hidden states, and let $I = \{i_1 < \cdots < i_m\} \subseteq \{1, \ldots, s\}$ be the indices selected by the splitting function, defining the new sequence length $m$ with $1 < m \leq s$. Let $W \in \mathbb{R}^{d_{\text{in}} \times d_{\text{out}}}$ be a learnable projection, where $d_{\text{in}}$ and $d_{\text{out}}$ correspond to the dimensions of the current and following stages, respectively. The pooled (downsampled) sequence is obtained by selecting the indexed rows and applying the linear projection:

$$Y = X_I W \in \mathbb{R}^{m \times d_{\text{out}}},$$

where $X_I = [X_{i_1}; \ldots; X_{i_m}]$.

**Upsampling.** The upsampling step maps coarse representations to finer ones for the next stage. As illustrated in Figure 2, we duplicate each coarse vector to match the length of the **following** segment, applying distinct, position-specific linear transformations to these duplicates.

Let $Y \in \mathbb{R}^{m \times d_{\text{out}}}$ and position-specific projections $\{W_p\}_{p=1}^K$ with $W_p \in \mathbb{R}^{d_{\text{out}} \times d_{\text{in}}}$. Given per-segment lengths $r_i \in \{1, \ldots, K\}$, compute

$$X_{(i,p)} = Y_i W_p \quad \text{for } i = 1, \ldots, m, p = 1, \ldots, r_i.$$

Stacking $(i, p)$ in segment-major order gives $X \in \mathbb{R}^{(\sum_{i=1}^m r_i) \times d_{\text{in}}}$.

Since these transformations are shared across segments but vary by position within a segment, we term this *Multi-Linear Upsampling*. In our experiments, models with multiple stages are more sensitive to the specific choice of upsampling strategy, whereas for pooling, many strategies work equally well.

### 2.1.2 Generation

During training, we process the entire input sequence in parallel, activating all stages simultaneously. At inference, generation is autoregressive: the byte-level stage is active at every step, while deeper stages activate less frequently according to the pooling pattern. Skip connections transmit information upward at each stage, so deeper stages can integrate fine-grained details. This cascading, conditional activation enables efficient inference: computationally intensive high-level stages activate rarely, but still effectively guide detailed lower-level predictions. In practice, this means that we need to cache the latest vector at the output of each stage to correctly propagate deeper stages' outputs.

### 2.2 Splitting Function

The AU-Net architecture supports flexible splitting strategies to define pooling points at each hierarchical stage. The primary constraint is that any chosen splitting function must be *stable to rightward insertion*: appending bytes should not alter prior pooling decisions, ensuring consistent autoregressive generation. Various methods (e.g., fixed windows [4], entropy [5], learned rules) are possible. Our current work splits on spaces using different regular expressions at each stage (details in Appendix B).

This strategy defines a hierarchy: Stage 1 processes raw bytes; Stage 2 pools at word boundaries (identified by the regex); Stage 3 pools after every two words (or sentence end); and Stage 4 after every four words (or sentence end). This rule-based approach, inspired by pre-tokenization in systems like GPT-4o's [7], is effective for Latin scripts. Extending robustly to languages without clear delimiters remains future work. Unlike prior approaches [5, 6, 8] that used similar splits mainly to replace BPE in a single-stage context, AU-Net uses these user-defined splits for its multi-stage hierarchical processing.

### 2.3 Evaluating on different scales

Large language models scale very predictably [9–11]. This allows us to estimate the performance of a model for a large compute budget. But more surprisingly, it allows us to predict the optimal hyperparameters for models way beyond our ablation budget. Bi et al. [11] described a method for sweeping learning rates (LR) and batch sizes (BSZ) across a range of small models, and they demonstrated that these results can be used to predict optimal hyperparameters for larger models.

Following their methodology, we show a different evolution of hyperparameters, both due to the data in our setup and to the hierarchical model. These hyperparameters are then used to do scaling laws for a bigger range of compute budgets to compare the baseline architecture and AU-Net. Throughout this paper, the *scale* of a run is its total pre-training compute $C$ measured in Floating Point Operation (FLOP):

$$C = \underbrace{F_{\text{model / input-unit}}}_{\text{FLOPs per (forward+backward) pass per input unit}} \times \underbrace{N_{\text{input-unit}}}_{\text{number of units of training input}} .$$

Following Bi et al. [11], we define model size as the number of FLOPs per input unit instead of relying on the number of parameters. This allows us to compare models with different architectures fairly. The formula for the number of FLOP per input-unit for a decoder-only transformer is given by:

$$F_{\text{model / input-unit}} = \underbrace{6N_{\text{params}}^{\text{no-embed}}}_{\text{linear term}} + \underbrace{6d\,L\,S}_{\text{attention term}} .$$

where, $N_{\text{params}}^{\text{no-embed}}$ is the number of parameters, excluding the embeddings. $d$ is the dimension, $S$ the sequence length and $L$ the number of layers. To scale up, one can either make the model bigger ($F_{\text{model / input-unit}} \uparrow$), give it more data ($N_{\text{input-unit}} \uparrow$), or do both. Gadre et al. [12] showed that keeping the *data-to-model ratio* $\gamma_{\text{input-unit}}$ constant is key to getting smooth scaling laws and predictable performance, where:

$$\gamma_{\text{input-unit}} = \frac{N_{\text{input-unit}}}{F_{\text{model / input-unit}}}.$$

We adopt this convention in all experiments and report the data-to-model ratio $\gamma_{\text{input-unit}}$ used in the experiments.

**Bytes versus tokens.** On DCLM [13], a token sequence is on average $k \approx 4.56$ times shorter than its byte sequence when using the LLaMa 3 tokenizer.

Given some compression factor $k$ between bytes and tokens, we want to express the equivalent $\gamma_{\text{bytes}}$. To do this, we note that $N_{\text{byte}} = k \times N_{\text{token}}$ and $F_{\text{model/byte}} = F_{\text{model/token}}/k$. Therefore,

$$\gamma_{\text{byte}} = k^2 \frac{N_{\text{token}}}{F_{\text{model/token}}} = k^2 \gamma_{\text{token}}.$$

Note that this scaling relationship is architecture-agnostic. The factor $\frac{1}{k}$ follows directly from the difference in sequence length between tokens and bytes under a given tokenizer. While different architectures may have distinct FLOPs/byte, the conversion between token- and byte-level compute is determined solely by the compression ratio $k$. This factor allows us to compare the performance of our model with the baseline on the same scale, as they will have seen the same amount of data and spent the same amount of FLOPs per token. Throughout the paper, we always express the data-to-model ratio in LLaMa 3 tokens ($\gamma_{\text{token}}$).

**FLOPS per byte for AU-Net.** In the case of AU-Net, we cannot use the same formula as the baseline because of the contraction and expansion happening in the model. However, we can still use the same formulas as long as we account for the contraction at each stage. So the total FLOPs per byte for AU-Net is simply the sum of each stage divided by the contraction factor.

$$F_{\text{model/byte}} = \sum_{i=1}^{L} \frac{F_{\text{model/byte}}^{i}}{k_i},$$

where $k_i$ is the contraction factor at stage $i$.

This property allows us to have models with a higher number of parameters for the same compute budget and data-to-model ratio.

**Hyperparameter scaling laws** Bi et al. [11] showed that the regularity of scaling laws can be exploited to tune very large models from a sweep over much smaller ones. We replicate their protocol on six miniature versions of each architecture (baseline Transformer and AU-Net): we perform a quasi-random search over batch size and learning rate, keep the configurations within 1% of the best validation loss, and fit $\text{BSZ}(C) = A\,C^{\alpha}$ and $\text{LR}(C) = B\,C^{\beta}$ to those points, with parameters $A, \alpha, B$ and $\beta$. We find the following formulas at the byte level for AU-Net:

$$\text{BSZ}_{\text{AU-Net}}(C) = 0.66 \times C^{0.321} \qquad \text{LR}_{\text{AU-Net}}(C) = 6.6 \times C^{-0.176}.$$

And we run the same tuning for the BPE baseline, for which we find:

$$\text{BSZ}_{\text{BPE}}(C) = 29.9 \times C^{0.231} \qquad \text{LR}_{\text{BPE}}(C) = 19.3 \times C^{-0.177}.$$

# 3 Experimental Results

## 3.1 Experimental Setup

**Data.** For all experiments, DCLM [13] served as the pretraining dataset, with a small portion held out for validation, totalling around 4T tokens (of GPTNeoXTokenizer). The corpus is mostly English and focuses on natural language understanding, i.e. it contains a marginal amount of code and maths.

**Baselines.** We compare our approach to different three baselines: Transformers equipped with the BPE tokenizer of LLaMa 3, Transformers and Mamba [14] trained directly on bytes, AU-Net variants using a fixed-size pooling window (denoted Transformer bytes[::$w$], where $w$ is the window size), similar to the Hourglass architecture [4]. To keep the comparison fair, we trained each baseline with the same amount of data or compute. For example, if a data budget of 273B bytes is used to train the bytes level or AU-Net model, this budget is converted to 60B training tokens for a transformer with LLaMa 3 tokenizer [15] because of the 4.56 compression rate measured on the DCLM corpus.

**AU-Net Architecture.** AU-Net-2, -3, and -4 progressively increase embedding dimensionality across stages ($512 \rightarrow 2048 \rightarrow 3072 \rightarrow 4608$ for deeper variants at the 1B scale), with layer allocation guided by the ablation results in appendix C.2. The AU-Net-2 architecture at 1B scale is illustrated in figure 1, showing stage dimensions, layer distribution, and total FLOP allocation per stage. The first byte-level stage does not require high dimensionality, as it primarily encodes local information before compression, whereas later stages capture increasingly abstract representations that benefit from greater capacity. Contraction rates are chosen so that when representations are merged, information density remains approximately constant (e.g., a sequence compressed by a factor of two doubles its hidden dimension). At the 1B scale, AU-Net-4 uses $6 + 25$ layers with a maximum hidden size of 4608, while the 8B-scale configuration expands to $6 + 33$ layers and a final hidden size of 4096. This setup reflects the model's compression rationale while keeping compute distribution flexible. For comparison, the BPE Transformer employs 25 layers with a hidden size of 2048 at the 1B scale and 33 layers with a hidden size of 4096 at the 8B scale.

Full architectural specifications, such as embedding sizes, layer counts, are detailed in appendix C.2. Also, note that, due to compute constraints, AU-Net-3 and AU-Net-4 were not trained at the 8B scale.

**Hyperparameters.** For a detailed overview of the hyperparameters, see appendix D. As explained in section 2.3, we sweep batch size and learning rate values across model scales ranging from 25M to 500M. Then, we extrapolate the best learning rate and batch size for any given compute budget.

**Evaluation Metrics.** All models are evaluated on a broad set of downstream tasks in a zero-shot setting, occasionally including a few in-context examples directly in the prompt. These tasks fall into two categories: (i) multiple-choice (MCQ) tasks, where the correct answer is selected as the option with the lowest normalized negative log-likelihood (divided by the number of characters) [16]; and (ii) open-ended generation tasks, where the model is allowed to freely generate its answer.

To highlight the strengths of AU-Net, we include specialized benchmarks targeting character-level manipulation (CUTE [17] appendix E) and low-resource language translation (FLORES-200, [18] section 3.4).

For clarity, we report a selection of key benchmark results in the main tables, including Hellaswag, ARC-Easy, ARC-Challenge, MMLU, NQ, TQA, and GSM8K. Also, we report 95% confidence intervals for all tables using bootstrap. A full breakdown of all evaluation results is provided in the appendix F.

In addition to task performance, the total training FLOPs and training throughput are provided for each model, measured in bytes per second per GPU (bps) on H100 80GB GPUs (internal cluster) during the actual training.

**Implementation Details.** As scaling is key to the success of large language models, our implementation balances efficiency and simplicity. We use *sequence packing* along with full attention, a strategy shown to have little to no impact on downstream performance ([13]). To reduce GPU memory pressure, all our experiments rely on Fully Sharded Data Parallelism (FSDP).

For additional speed-ups, the entire model is compiled with `torch.compile`. Compilation, however, requires a static computation graph, which clashes with the variable-length outputs produced by our adaptive pooling: the number of bytes per word (and thus per stage) naturally varies across sentences. We resolve this by fixing a maximum sequence length at every stage: sequences that exceed the limit are truncated abruptly, and shorter ones are padded. This compromise yields a graph that is static for compilation while still supporting adaptive hierarchical pooling in practice.

Table 2: **Downstream results comparing AU-Net to BPE and byte-level baselines.** We report accuracy on key benchmarks with 95% confidence intervals where applicable. Literature models are shown in *italics*; all models are trained on the same corpus, unless specified. AU-Net variants differ in the number of stages. We also report compute budget and empirical training speeds in bytes/sec.

| Model | Params | Emb. | Flops | bps | Hellaswag | ARC_E | ARC_C | MMLU | NQ | TQA | GSM8k |
|---|---|---|---|---|---|---|---|---|---|---|---|
| **Dim=2048 (1B model), 60B tokens (data-to-model ratio of 10)** | | | | | | | | | | | |
| Mamba Byte | 1.3B | 1M | 3e21 | 32k | 63.0 $\pm0.9$ | 60.3 $\pm2.0$ | 33.6 $\pm2.8$ | 25.1 $\pm0.7$ | 8.2 $\pm0.9$ | 21.2 $\pm0.7$ | 2.1 $\pm0.8$ |
| Transformer Byte | 1.3B | 1M | 4e21 | 47k | 63.0 $\pm1.0$ | 61.2 $\pm1.9$ | 34.7 $\pm2.7$ | 24.7 $\pm0.7$ | 8.8 $\pm0.9$ | 21.4 $\pm0.8$ | 2.5 $\pm0.9$ |
| Transformer Byte[::4] | 1.3B | 1M | 6e20 | 180k | 63.5 $\pm0.9$ | 63.0 $\pm1.9$ | 36.0 $\pm2.7$ | 25.1 $\pm0.7$ | 6.8 $\pm0.8$ | 17.2 $\pm0.7$ | 2.6 $\pm0.9$ |
| Transformer Byte[::5] | 1.3B | 1M | 5e20 | 218k | 60.0 $\pm1.0$ | 60.0 $\pm2.0$ | 34.8 $\pm2.8$ | 23.9 $\pm0.7$ | 5.6 $\pm0.7$ | 14.7 $\pm0.7$ | 2.0 $\pm0.8$ |
| Transformer Byte[::6] | 1.3B | 1M | 4e20 | 255k | 58.6 $\pm1.0$ | 59.5 $\pm2.0$ | 32.4 $\pm2.7$ | 24.9 $\pm0.7$ | 4.6 $\pm0.7$ | 12.3 $\pm0.6$ | 2.3 $\pm0.8$ |
| AU-Net 2 | 1.3B | 1M | 5e20 | 225k | 64.2 $\pm0.9$ | 64.4 $\pm1.9$ | 35.2 $\pm2.8$ | 24.8 $\pm0.7$ | 8.8 $\pm0.9$ | 20.4 $\pm0.7$ | 2.7 $\pm0.9$ |
| AU-Net 3 | 2.5B | 1M | 7e20 | 180k | **67.4** $\pm0.9$ | 65.9 $\pm1.9$ | 36.7 $\pm2.7$ | **26.3** $\pm0.7$ | **9.6** $\pm1.0$ | 22.6 $\pm0.8$ | 2.3 $\pm0.8$ |
| AU-Net 4 | 4.2B | 1M | 8e20 | 155k | 66.4 $\pm0.9$ | **67.4** $\pm1.9$ | **37.0** $\pm2.8$ | **26.3** $\pm0.7$ | 5.1 $\pm0.7$ | 15.5 $\pm0.7$ | **3.5** $\pm1.0$ |
| Transformer BPE | 1.8B | 525M | 7e20 | 210k | 63.6 $\pm1.$ | 62.8 $\pm1.$ | 36.5 $\pm2.$ | 26.2 $\pm0.$ | 8.8 $\pm0.9$ | **26.3** $\pm0.$ | 2.3 $\pm0.8$ |
| **Dim=2048 (1B model), 370B tokens (data-to-model ratio of 40)** | | | | | | | | | | | |
| AU-Net 2 | 1.3B | 1M | 3e21 | 225k | 69.9 $\pm0.9$ | 68.6 $\pm1.9$ | 38.9 $\pm2.7$ | 28.8 $\pm0.7$ | 13.0 $\pm1.1$ | 32.5 $\pm0.9$ | 3.0 $\pm0.9$ |
| AU-Net 3 | 2.5B | 1M | 4e21 | 180k | 72.9 $\pm0.9$ | 72.3 $\pm1.8$ | **43.3** $\pm2.8$ | 28.0 $\pm0.7$ | **15.3** $\pm1.2$ | **39.1** $\pm0.9$ | 3.7 $\pm1.0$ |
| AU-Net 4 | 4.2B | 1M | 5e21 | 155k | **73.7** $\pm0.9$ | **72.6** $\pm1.8$ | 43.2 $\pm2.9$ | **31.7** $\pm0.7$ | 14.0 $\pm1.1$ | 35.5 $\pm0.9$ | **5.3** $\pm1.2$ |
| Transformer BPE | 1.8B | 525M | 4e21 | 210k | 70.2 $\pm0.$ | 68.6 $\pm1.$ | 38.5 $\pm2.$ | 27.0 $\pm0.$ | 13.5 $\pm1.$ | 37.2 $\pm0.$ | 4.4 $\pm1.1$ |
| *DCLM-1B-5×(145B)*[13] | 1B | 207M | 1e21 | - | 66.1 | 70.2 | 40.6 | 26.4 | - | 29.3 | 1.1 |
| *MegaByte (263B)*[19] | 1.1B | - | - | 73k | 38.9 | 54.9 | 23.4 | 25.1 | - | 9.6 | |
| *Hierarchical (263B)*[6] | 1.1B | - | 1e21 | - | 46.5 | 65.0 | 30.5 | 26.0 | - | 9.6 | - |
| **Dim=4096 (8B model), 200B tokens (data-to-model ratio of 5)** | | | | | | | | | | | |
| AU-Net 2 | 7.9B | 1M | 1e22 | 41k | **79.1** $\pm0.8$ | **80.0** $\pm1.6$ | **51.2** $\pm2.9$ | **51.1** $\pm0.8$ | **22.1** $\pm1.3$ | 50.9 $\pm0.9$ | 10.0 $\pm1.6$ |
| Transformer BPE | 7.5B | 1B | 9e21 | 43k | 77.2 $\pm0.$ | 74.5 $\pm1.$ | 49.2 $\pm2.$ | 49.6 $\pm0.$ | 21.1 $\pm1.$ | 51.1 $\pm0.$ | **10.7** $\pm1.7$ |
| *DCLM-7B-2×(276B)*[13] | 7B | 413M | 1e22 | - | 77.8 | 78.1 | **52.6** | 50.8 | - | **50.9** | 4.3 |
| *Hierarchical (263B)*[6] | 9.2B | - | 1e22 | 15k | 56.3 | 76.6 | 44.2 | 32.0 | - | 33.1 | - |
| *BLT (220B)$^×$*[5] | 8B | - | 1e22 | - | 72.2 | 66.8 | 38.8 | 25.2 | - | - | - |
| *BLT (1T)$^*$*[5] | 8B | - | 5e22 | - | 80.6 | 79.6 | 52.1 | 57.4 | - | - | - |
| *DCLM-7B (2.5T)$^*$*[15] | 7B | 413M | 1e23 | - | 80.4 | 82.2 | 59.9 | 63.7 | - | 52.7 | 2.5 |
| *LLaMa 3.1 (15T)$^×$*[15] | 8B | 1B | 6e23 | - | 83.3 $\pm0.8$ | 80.7 $\pm1.5$ | 54.8 $\pm2.9$ | 66.4 $\pm0.8$ | 29.1 $\pm1.5$ | 64.4 $\pm0.9$ | 54.7 $\pm2.7$ |

$^*$ Trained on mix of DCLM and other datasets

$^×$ Trained on different corpus than DCLM

## 3.2 Equal Data Budget Results

We evaluate the effectiveness of hierarchical pooling by fixing the model's primary hidden dimension to 2048 and maintaining a constant total training-data budget. The hidden dimension at each stage is scaled proportionally to its contraction ratio as described in section 2.1. For instance, the byte-level stage uses a dimension of $2048/4 = 512$, the word-level stage uses 2048, and the 2-word level uses $1.5 \times 2048 = 3072$, continuing in this manner for deeper stages. We assess the downstream performance of language models with 2, 3, and 4 stages at the 1B parameter scale. For the 8B model, we evaluate only the 1-stage configuration for now. All variants are compared against a Transformer baseline using the LLaMA 3 tokenizer of the same main hidden dimension. More ablations regarding pooling and the number of layers per stage can be found in the appendix C.

As shown in table 2, hierarchical models consistently match or outperform their BPE-based counterparts. This trend holds across various configurations and becomes especially pronounced as we introduce more hierarchical stages. Notably, multi-stage AU-Net models (e.g., AU-Net 3 and AU-Net 4) outperform BPE baselines on several benchmarks. An interesting exception to this pattern is the TQA benchmark, which is a knowledge-intensive task evaluating the generation of the model. AU-Net models along with byte-level baselines consistently underperform on TQA compared to BPE-based models. This suggests that the performance gap may not stem solely from the hierarchical structure. However, as model size and training data scale (e.g., at the 8B or 1B, 370B tokens scale), this discrepancy seems to vanish. When examining the Transformer Bytes[::$w$] models, we clearly observe the effect of pooling at word boundaries. As the pooling window size $w$ increases, the resulting increasing compression comes at the cost of a consistent drop in performance. In contrast, using a more principled pooling strategy, as in AU-Net, achieves better compression while main-

taining strong downstream performance, enabling more robust and faster training. This effect is particularly pronounced on generative tasks such as TQA, where performance declines sharply for larger window sizes and remains below both AU-Net and BPE baselines. This is further emphasised in the subsequent section 3.4 where the same pattern emerges. For languages where space-based pooling does not align well with word boundaries, performance also decreases significantly.

We observe early signs of diminishing returns beyond a certain number of stages. While AU-Net 4 improves on reasoning-heavy tasks such as ARC-C and GSM8k, gains on benchmarks like Hellaswag and TQA are less consistent. However, this effect may stem not from hierarchy itself, but from data efficiency: deeper hierarchies might require more training data to reach their full potential. Supporting this interpretation, AU-Net 2 and AU-Net 4 benefit significantly from additional training data, and that MMLU and GSM8k scores continue to improve with more stage, even at fixed scale.

Finally, when comparing our models to similarly sized baselines from the literature (italicized in the table), we find that AU-Net remains competitive, even while using significantly less training data. For instance, BLT (1T) uses approximately 5× more compute than our 8B model, while only being better on MMLU. Importantly, comparisons with literature models are fair, as all were trained on the same corpus: DCLM (except for BLT (220B) and LLAMA 3.1 (15T)).

To further evaluate our approach, we now turn to scaling laws (figure 3) to better quantify how our architecture compares to a standard Transformer with BPE. We focus on AU-Net 2 and AU-Net 3, using a data-to-model ratio of 2. This choice is motivated by the diminishing returns observed when moving from AU-Net 3 to AU-Net 4 under the same data-to-model ratio.

### 3.3 Scaling laws

Using the learning rate and batch size formulas (Section 2.3), we run pretraining for a range of compute budgets ranging from 1e19 to 1e22 flops (corresponding to models from 150M to 5.3B non embedding parameters) for the baseline, with a data-to-model ratio of 10. This is roughly 2× the optimal data-to-model ratio found by Kaplan et al. [9]. Compared to table 2 and the 8B-scale experiments, the scaling-law runs use a higher data-to-model ratio (data-to-model ratio = 10 instead

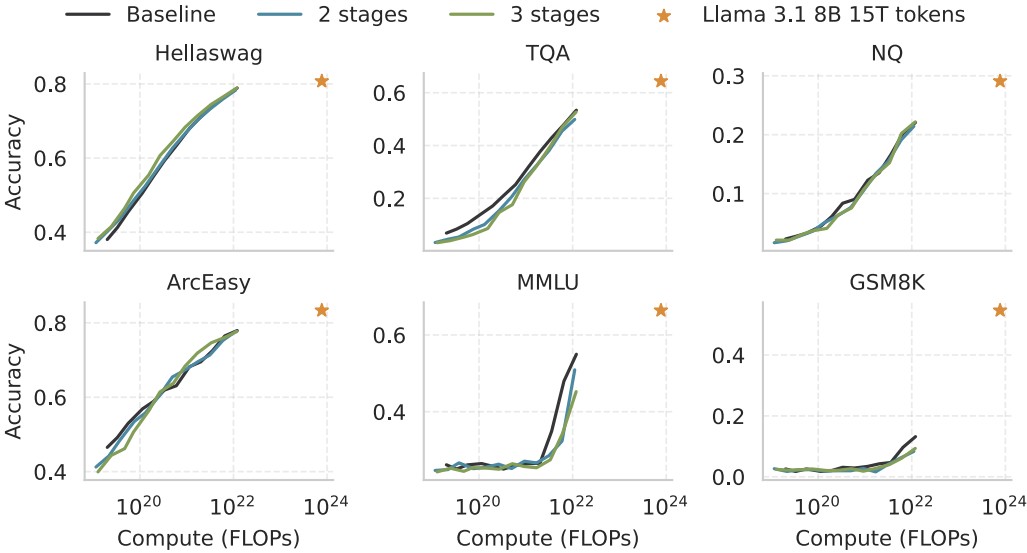

Figure 3: Downstream task performance scaling with compute (1e19-1e22 FLOPs) under data-to-model ratio of 10. AU-Net (2/3 stages) generally tracks a strong BPE Transformer baseline, which itself performs competitively against much larger models (e.g., LLaMa 3.1 8B on 15T tokens 100× compute). While AU-Net matches the baseline on tasks like Hellaswag and ARC Easy, and catches up on TQA at higher compute, its performance improvement phase on MMLU and GSM8K appears to start later. The general underperformance on GSM8K is also linked to limited math data in the DCLM pretraining corpus.

Table 3: Multilingual evaluation. **Left:** BLEU scores on the FLORES-200 benchmark across multiple languages. Higher scores indicate better translation quality. **Right:** MMLU Exact Match (%) across 26 non-English languages. Results are averaged per language across all tasks.

| FLORES-200 (BLEU) | Lang. → Eng. | | Eng. → Lang. | |
|---|---|---|---|---|
| | BPE | AU-Net 2 | BPE | AU-Net 2 |
| German | **34.4**±1.2 | 33.9±1.2 | **16.7**±0.8 | 15.6±0.7 |
| Dutch | 24.7±1.0 | **25.0**±1.0 | **12.3**±0.6 | 11.7±0.6 |
| Afrikaans | 32.0±1.3 | **35.7**±1.3 | 14.8±0.8 | **16.1**±0.8 |
| Faroese | 8.7±0.7 | **9.9**±0.8 | 1.8±0.3 | **2.9**±0.4 |
| Icelandic | 7.8±0.6 | **9.0**±0.7 | 1.7±0.3 | **2.5**±0.3 |
| Limburgish | 15.3±0.9 | **19.9**±1.0 | 5.7±0.4 | **6.7**±0.5 |
| Luxembourgish | 11.4±0.8 | **14.7**±0.9 | 2.6±0.3 | **4.0**±0.3 |
| Italian | 29.1±1.0 | **30.1**±1.0 | 15.1±0.7 | **15.3**±0.6 |
| Friulian | 14.6±0.8 | **19.1**±1.0 | 3.2±0.3 | **4.0**±0.3 |
| Ligurian | 16.5±0.9 | **21.8**±1.0 | 3.4±0.3 | **3.9**±0.3 |
| Lombard | 12.9±0.9 | **19.2**±1.0 | **5.2**±0.4 | 4.2±0.3 |
| Sardinian | 14.3±0.8 | **18.2**±1.0 | 4.3±0.4 | **4.5**±0.4 |
| Sicilian | 11.7±0.8 | **16.8**±0.9 | 3.9±0.4 | **4.7**±0.4 |
| Venetian | 19.8±1.0 | **25.4**±1.1 | **5.8**±0.4 | 5.6±0.4 |
| Spanish | 28.2±1.0 | **29.3**±1.0 | **20.2**±0.7 | 19.8±0.7 |
| Asturian | 24.0±1.1 | **28.6**±1.1 | **10.3**±0.6 | 8.2±0.5 |
| Catalan | 28.1±1.1 | **33.0**±1.2 | 9.6±0.5 | **10.7**±0.6 |
| Occitan | 28.0±1.2 | **35.5**±1.2 | 4.8±0.4 | **6.2**±0.4 |
| Portuguese | 42.0±1.3 | **43.6**±1.3 | 25.3±1.0 | **25.4**±1.0 |
| Galician | 29.6±1.1 | **34.0**±1.2 | 9.9±0.5 | **10.2**±0.6 |
| Papiamento | 17.3±0.9 | **22.1**±1.1 | 2.5±0.3 | **6.3**±0.4 |
| Kabuverdianu | 13.7±0.9 | **20.8**±1.1 | 2.4±0.3 | **5.1**±0.4 |
| Esperanto | 15.9±1.0 | **19.3**±1.0 | 3.6±0.4 | **5.9**±0.4 |
| **Average** | 20.9±0.2 | **24.6**±0.2 | 8.0±0.1 | **8.7**±0.1 |

| MMLU | BPE | AU-Net 2 |
|---|---|---|
| English | 49.6±0.8 | **51.1**±0.8 |
| Arabic | 29.1±0.8 | **29.5**±0.8 |
| Bengali | 27.5±0.7 | **27.6**±0.8 |
| Chinese | **33.0**±0.8 | 28.0±0.7 |
| Czech | 30.7±0.8 | **32.2**±0.8 |
| Dutch | 34.5±0.8 | **37.1**±0.8 |
| Finnish | 29.0±0.7 | **29.3**±0.7 |
| French | 37.3±0.8 | **40.7**±0.8 |
| German | 36.0±0.8 | **37.6**±0.8 |
| Greek | 29.2±0.8 | **30.5**±0.8 |
| Hindi | **27.9**±0.7 | 27.5±0.7 |
| Hungarian | 29.0±0.8 | **30.1**±0.8 |
| Indonesian | 34.9±0.8 | **37.3**±0.8 |
| Italian | 36.2±0.8 | **39.0**±0.8 |
| Japanese | **29.5**±0.7 | 28.2±0.7 |
| Korean | **28.4**±0.7 | 28.2±0.8 |
| Persian | **28.7**±0.7 | 28.6±0.7 |
| Polish | 30.3±0.8 | **32.0**±0.8 |
| Portuguese | 37.2±0.8 | **40.9**±0.8 |
| Romanian | 34.0±0.8 | **36.9**±0.8 |
| Russian | 30.9±0.8 | **31.2**±0.8 |
| Spanish | 37.6±0.8 | **41.4**±0.8 |
| Swahili | 28.8±0.7 | **29.9**±0.8 |
| Swedish | 33.5±0.8 | **36.0**±0.8 |
| Telugu | 26.8±0.7 | **27.4**±0.7 |
| Thai | **28.0**±0.7 | 27.5±0.7 |
| Turkish | 29.1±0.7 | **30.0**±0.7 |
| Vietnamese | **31.4**±0.8 | 30.7±0.7 |
| **Average** | 31.4±0.1 | **32.4**±0.1 |

of 5) and the latest hyperparameters described at the end of section 2.3. These differences can explain the variation in peak performance across the two experiments.

The list of models chosen for each budget is detailed in the appendix G. Figure 3 shows the evolution of performance on 6 downstream tasks for AU-Net and the BPE baseline. Here we mainly notice that 2 and 3 stage AU-Net models can match the performance of the BPE baseline when carefully controlling for compute budget. This is the case for Hellaswag, Arc Easy, and NQ. For TQA, AU-Net both for 2 and 3 stages starts with a performance gap, but the 3 stage model catches up at 1e22 flops. However, both 2-stage and 3-stage AU-Net models are still behind the BPE baseline at 1e22 flops for GSM8K and MMLU. Most downstream tasks follow a sigmoid pattern: performance is near chance at low compute, then rapidly improves before plateauing. For AU-Net models, this transition appears to occur slightly later on tasks like GSM8K and MMLU, suggesting that the benefits of a deep hierarchy may become more pronounced at larger scales. Nevertheless, on many benchmarks, both our AU-Net variants and our BPE baseline achieve results remarkably close to those of considerably larger models like LLaMa 3.1 8B (pretrained on 15T tokens, representing 100 times more compute than our largest run shown here). This proximity underscores the strength of our BPE baseline, making AU-Net's ability to match or trend towards it particularly noteworthy. The primary exception where this close tracking is less apparent is GSM8K; however, this underperformance across all our models is likely due to the pretraining corpus, as DCLM contains very little math data.

## 3.4 Extended Evaluations

We present results highlighting two specific advantages of byte-level training with AU-Net over BPE-based Transformers: improved performance on multilingual benchmarks (Table 3) and character-level manipulation tasks (Table 7 in the appendix E). Table 3 show surprisingly strong non-English performance of both model, despite DCLM being heavily filtered toward English.

**Cross-lingual generalization within language families.** On the multilingual MMLU benchmark (Table 3 right), languages using Latin scripts consistently benefit from byte-level modeling. We observe strong positive transfer between related languages. Concretely, Germanic languages (German, Swedish, Dutch, etc.) show an average gain of +3.0 points, while Romance languages (Italian, Spanish,

Portuguese, French, etc.) improve by +4.0 points. These results suggest that operating at the byte level helps the model to capture shared orthographic and morphological patterns across related languages.

**Transfer to low-resource languages.** The FLORES-200 benchmark (Table 3 left) includes many low-resource languages that are underrepresented or absent in the training data. This setting allows us to test the model's ability to generalize based on subword morphology and shared linguistic roots. Byte-level modeling provides the flexibility to construct meaningful representations without requiring the presence of these languages in the tokenizer or training corpus. We observe consistent gains in translation tasks into English, where the model must primarily understand the source language. The advantage is particularly clear for languages that share syntactic or morphological traits with more dominant relatives in the same family. This also highlights the robustness of our model: it can produce meaningful translations even with out-of-vocabulary words or forms unseen during training. In the reverse direction (English to low-resource), generation remains more challenging.

## 4   Related Work

Traditional tokenization methods are important for computational efficiency [20–23], but impose fixed granularities. Early attempts to overcome this rigidity explored adaptive vocabularies [24], n-gram combinations [25], or alternative splitting criteria like entropy [5]. Our work, AU-Net, advances this by integrating tokenization and representation learning into a multi-level, autoregressive U-Net architecture that operates directly on bytes.

This hierarchical, adaptive-pooling design distinguishes AU-Net from prior works. For instance, Megabytes [19] introduce a two stage LLM using local models but with fixed-size token blocks, unlike AU-Net's input-adaptive pooling. Neitemeier et al. [6], Byte Latent Transformers (BLT) [5], Dynamic Pooling Transformer (DPT) [26] and SpaceByte [8] also process bytes or use specialized splitting functions. However, they typically aim to replace BPE for a single effective processing stage or use local attention mechanisms. In contrast, AU-Net leverages user-defined splits within a multi-stage architecture featuring distinct pooling strategies that differ from the cross-attention methods in Nawrot et al. [4], Pagnoni et al. [5]. Nawrot et al. [4] defined a similar U-Net architecture but with fixed pooling, much smaller models, and their evaluations mainly focus on perplexity.

Concurrent to our work, H-Net [27] introduces a hierarchical sequence model based on learnable, data-dependent dynamic chunking. Like AU-Net, H-Net operates directly on bytes and constructs up to three stages network. While both models share similar high-level goals, they differ in key mechanisms: H-Net employs a learned dynamic split function, whereas AU-Net relies on predefined, rule-based splitting functions for hierarchical pooling and upsampling.

## 5   Conclusion

This paper introduces AU-Net, an autoregressive U-Net that processes raw bytes and learns hierarchical token representations. By dynamically pooling bytes into words and multi-word chunks, AU-Net eliminates the need for predefined vocabularies and large embedding tables while preserving BPE performance with higher compression. Experiments show that AU-Net matches strong BPE baselines under controlled compute budgets, with deeper hierarchies demonstrating promising scaling trends. Furthermore, its byte-level operation leads to improved performance on character-level tasks and better generalization to low-resource languages. This approach offers a flexible and efficient alternative to traditional tokenization methods, paving the way for more adaptable and versatile language models.

**Limitations and further work**

Our work relies on DCLM, an English-only corpus, and currently supports only space-delimited languages with a predefined splitting function. For example, this affects, Chinese, where MMLU scores are lower than the BPE baseline. Another limitation is the lack of evaluation on code and math benchmarks. Since the DCLM corpus contains very little code abd mathematical data, models trained on it perform poorly on these tasks, yielding low and noisy results close to random performance. One extension could be to learn directly the splitting function. On the software side, as the number of parameters increases with the number of stages, FSDP already struggles to overlap computation and communication even at 3/4 stages, it needs a minimum amount of inputs to be fully overlapped.

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
