# OpenReview forum: "From Bytes to Ideas: Language Modeling with Autoregressive U-Nets"
_NeurIPS.cc/2025/Conference — NeurIPS 2025 poster_

### Official Review · Reviewer_T1Se · 2025-06-19

**Clarity:** 3
**Significance:** 2
**Originality:** 2
**Rating:** 2
**Confidence:** 4

**Summary:**

The paper introduces AU-Nets, a family of hierarchical language models intended to enable byte-level language modelling.

In contrast to some prior hierarchical architectures [[1], [2]], the authors adaptively chunk the input bytes based on a Regex instead of using fixed-length chunks and generalize the architecture beyond a two-stage hierarchy (AU-Net-2) to higher-level hierarchies (AU-Net-3 and AU-Net-4).

Experiments on pretraining AU-Net models up to 8B parameters show that they perform roughly on par or slightly worse than models using subword tokenization trained in a compute-matched way.

[1]: https://arxiv.org/abs/2110.13711
[2]: https://arxiv.org/abs/2305.07185

**Questions:**

See Weaknesses.

**Ethical Concerns:**

["NO or VERY MINOR ethics concerns only"]

**Final Justification:**

I am maintaining my recommendation to Reject since:

- Per the authors, the main contribution of the paper is showing 'that going deeper with hierarchical architectures can work under the right circumstances and has advantages'. I do not think this is sufficiently well supported. (1) we already know from prior work (e.g. [[1]]) that a two-stage architecture can be beneficial and (2) the authors' results for going beyond two stages seem discouraging to me, in particular in Table 2 we see that models with three or four stages perform at best slightly better at vastly increased parameter counts, so if we want improved performance at matched FLOPs it seems like a better choice to use a MoE instead of a three or four stage hierarchy (a comparison the paper doesn't make).
- The auxiliary contributions (faster inference, improved performance over BLT) either lack empirical evidence or are confounded by spurious factors such as different training data.

[1]: https://arxiv.org/abs/2412.09871

**Limitations:**

yes

**Quality:**

2

**Strengths And Weaknesses:**

Strengths:
- Research on direct byte-level language modelling is important due to the many inadequacies of subword tokenization and potentially highly impactful.
- Experiments are comprehensive and have a strong baseline.
- The paper is mostly well written and easy to follow.

Weaknesses:
- Although scaling the levels of the hierarchy (AU-Net-3 and AU-Net-4) slightly improves performance in a FLOP-matched training setup, it seems like a problematic dimension to scale due to the vastly increased parameter count (1.3B for AU-Net-2 vs. 4.2B for AU-Net-4 at slightly improved task performance).
- It is not clear how AU-Net-2 differs from prior work using adaptive pooling heuristics, e.g. the whitespace-split setting of BLT [[3]] and DTP [[4]] (DTP should also be referenced). BLT is only shown to be worse empirically, but there is no reason given as to why this may be the case. There is also no controlled comparison against these prior byte-level architectures. Is the only substantial difference the Regex splitting (vs. whitespace)? Or could the superior performance of AU-Nets be caused by other spurious factors related to differences in the training setup?

[3]: https://arxiv.org/abs/2412.09871
[4]: https://arxiv.org/abs/2211.09761

---

> ### Author Rebuttal · Authors · 2025-07-30
>
> We thank the reviewer for this constructive review, and answer below the questions raised.
>
>
> ---
>
>
> > **Although scaling the levels of the hierarchy (AU‑Net‑3/4) slightly improves performance in a FLOP‑matched setup, it seems like a problematic dimension to scale due to the vastly increased parameter count (1.3B for AU‑Net‑2 vs. 4.2B for AU‑Net‑4) at only slightly improved task performance.**
>
> You are right that adding stages increases the number of parameters and therefore GPU memory and checkpoint/storage size. However, it comes with other advantages, explained below.
>
> - Because we match FLOPs per byte to the baseline, the training and inference compute per generated byte are the same. Extra parameters live mostly in higher stages that process shorter, downsampled sequences, so they add capacity without increasing per‑byte compute.
>
> - **Inference can be faster** at a fixed quality target because the total KV cache can be smaller: upper stages cache over much shorter sequences, which reduces aggregate KV memory/traffic and can improve inference time in practice. (Inference is memory-bound)
>
> - AU-Net decouples parameters from compute cost. A flat Transformer’s cost is often approximated as ≈ 6 × (#params) × (#tokens). In AU‑Nets, this is no longer the case; by varying depth, one can choose a **larger‑parameter model at the same per‑byte cost** (assuming GPU memory is not a big issue).
>
> - **Pooling extends effective context**: deeper hierarchies handle longer inputs at the same compute budget and can predict further ahead. We do not claim definitive wins on such long‑horizon tasks yet; our benchmarks are mostly “classic LLM” and underweight long context and reasoning/generation. We plan to apply AU‑Nets to code/math settings where this matters more in the future.
>
> - Finally, in **high data‑to‑model (“overtrained”) regimes**, we observe a **positive trend**: deeper AU‑Nets behave better, with AU‑Net‑4 competitive with or slightly better than AU‑Net‑2/3. This is encouraging since most, if not all, state-of-the-art models are heavily overtrained.
>
>
> > **It is not clear how AU‑Net‑2 differs from prior work using adaptive pooling heuristics, e.g., BLT (whitespace split) and DTP (which should be referenced). BLT is only shown to be worse empirically, without a reason. No controlled comparison. Is Regex splitting the only substantial difference, or could gains be from training‑setup differences?**
>
> We should be clearer in explaining that regex splitting is not the only difference. AU‑Nets differ along two architectural axes and one optimization axis:
> - Contextual downsampling: BLT embeds each patch/word independently (local pooling), whereas AU‑Nets use sliding‑window attention to form higher‑level tokens, allowing cross‑boundary context before downsampling and reducing boundary‑induced information loss.
> - Simple, stable upsampling: AU‑Nets use linear upsampling without attention, which is more stable in byte‑level training in our experience.
> - A practical contribution of our work is byte‑level–specific optimization guidance. Stable, high‑quality byte‑level pretraining benefits from different scaling for learning rate and batch size than subword models (e.g., more conservative LR and larger global batch/warmup). We consider this to be an important finding that contributes to having a model competitive with BPE.
>
> On comparisons: we compute‑match pretraining, train a strong byte‑level baseline under the same optimizer/schedule family, and report results alongside published BLT and Hierarchical‑Transformer, both of which used DCLM as part of their pretraining data. While we did not re‑train BLT/DTP/Hierarchical‑Transformer under our hyperparameters due to compute constraints, using similar pretraining data provides a fair and informative comparison. We will add the missing DTP citation; to our knowledge, DTP does not report directly comparable downstream evaluations on any benchmark.

---

> ### Comment · Reviewer_T1Se · 2025-08-01
>
> Thank you for your response. About the two points:
>
> ### Vastly increased parameter count at slightly increased task performance
>
> > Inference can be faster at a fixed quality target because the total KV cache can be smaller
>
> Faster inference is indeed attractive, but if this is a central contribution it would need supporting experiments and discussion in the paper. It is hard to trust this claim with zero empirical evidence.
>
> > one can choose a larger‑parameter model at the same per‑byte cost (assuming GPU memory is not a big issue)
>
> This achieves the same effect as a mixture of experts, but from the only slightly improved scores from more AU-Net hierarchies in the paper it seems like mixtures-of-experts might scale more favorably. To make this claim effectively, the paper would need to compare against MoEs in some way.
>
> > Pooling extends effective context: deeper hierarchies handle longer inputs at the same compute budget and can predict further ahead. We do not claim definitive wins on such long‑horizon tasks yet; our benchmarks are mostly “classic LLM” and underweight long context and reasoning/generation. We plan to apply AU‑Nets to code/math settings where this matters more in the future.
>
> I would be very curious to see these results!
>
> ### Comparison to prior methods (primarily BLT)
>
> Regarding the differences to BLT (which could non-spuriously cause the high score difference):
>
> > Contextual downsampling: BLT embeds each patch/word independently (local pooling), whereas AU‑Nets use sliding‑window attention to form higher‑level tokens, allowing cross‑boundary context before downsampling and reducing boundary‑induced information loss.
>
> This is not correct. BLT uses a sliding window of 512 bytes (across patch boundaries) in the local encoder in the released checkpoints: https://github.com/facebookresearch/blt/blob/4ae7a625940743c9438a20ee8a8d7fab898bcd69/bytelatent/model/local_models.py#L258-L285. In this sense, the two are equivalent.
>
> > Simple, stable upsampling: AU‑Nets use linear upsampling without attention, which is more stable in byte‑level training in our experience.
>  > A practical contribution of our work is byte‑level–specific optimization guidance. Stable, high‑quality byte‑level pretraining benefits from different scaling for learning rate and batch size than subword models (e.g., more conservative LR and larger global batch/warmup). We consider this to be an important finding that contributes to having a model competitive with BPE.
>
> The finding on upsampling is interesting but would need experimental support via an ablation.
>
> Overall, the response does not address my main concern. From the paper, it seems like AU-Nets vastly outperform BLT with similar FLOPs/token budgets. My question is: where is this coming from? And are these differences really due to the AU-Net architecture or due to tricks which when applied to BLT in the same way would yield equivalent or better performance? It seems like it might be the latter.
>
> You mentioned two differences (I am skipping the first one since I believe it is incorrect): the upsampling and the learning rate schedule / batch size. Both of these would need to be ablated and compared with BLT to properly show that AU-Net's are indeed the better choice.
>
> I am thus maintaining my score.

---

> > ### Author Response · Authors · 2025-08-05
> >
> > Thank you for your detailed response.
> >
> > ---
> > > **This is not correct. BLT uses a sliding window of 512 bytes (across patch boundaries) in the local encoder in the released checkpoints: https://github.com/facebookresearch/blt/blob/4ae7a625940743c9438a20ee8a8d7fab898bcd69/bytelatent/model/local_models.py#L258-L285. In this sense, the two are equivalent.**
> >
> > We sincerely apologize for misunderstanding what BLT does for the local model. We will correct this and try to be more precise in our comparisons.
> >
> > > **Faster inference is indeed attractive, but if this is a central contribution, it would need supporting experiments and discussion in the paper. It is hard to trust this claim with zero empirical evidence.**
> >
> > > **To make this claim effectively, the paper would need to compare against MoEs in some way.**
> >
> > You are right that inference and comparison with MoE needs empirical evidence, and we would love a chance to do that in future work that builds on this one. We wanted to limit the scope of the paper to a few solid claims, which are that going deeper with hierarchical architectures can work under the right circumstances and has advantages (this builds on BLT instead of replacing it).
> >
> > > **Overall, the response does not address my main concern. From the paper, it seems like AU-Nets vastly outperform BLT with similar FLOPs/token budgets. My question is: where is this coming from? And are these differences really due to the AU-Net architecture or due to tricks which when applied to BLT in the same way would yield equivalent or better performance? It seems like it might be the latter.**
> >
> > We compare to a very strong baseline and make the case that hierarchical models with two stages (similar to BLT) or multiple stages are competitive with BPE-based models. We do not claim that BLT’s architecture is fundamentally worse, but we show that we found some key changes to the 2-stage setting that give much better results, and show promising results for deeper hierarchies. We unfortunately cannot cover everything (inference, comparison to MOE, rerunning BLT, DTP on the same setup) but would love to cover those in future work as they represent important and interesting comparisons deserving dedicated study. We don’t want to mischaracterize or be unfair to BLT or other related work, so we can highlight the architectural similarities in the paper and point to the training setup for the differences there. However, the contributions of the paper are not limited to those differences; we believe that our main comparison point is against the strong BPE baseline we produced, as it is the most comparable.

---

> > > ### Comment · Reviewer_T1Se · 2025-08-06
> > >
> > > > We unfortunately cannot cover everything (inference, comparison to MOE, rerunning BLT, DTP on the same setup) but would love to cover those in future work as they represent important and interesting comparisons deserving dedicated study.
> > >
> > > I understand that it is hard to cover all of this, and apologize if my response came across as necessarily demanding all these experiments. Instead, my concerns can be summarized as follows:
> > >
> > > 1. Your claims need supporting evidence. In your rebuttal, you claimed faster inference and a 'a fair and informative comparison' to BLT, both of which lack empirical support as per our discussion above.
> > > 2. The claim in your latest response 'that going deeper with hierarchical architectures can work under the right circumstances and has advantages' is indeed more important as is it the one your paper focuses the most on. However, as I mentioned above I am not convinced by it since (1) we already know from prior work that a two-stage architecture can be beneficial and (2) your results for going beyond two stages seem somewhat discouraging to me, in particular in Table 2 we see that models with three or four stages perform at best slightly better at vastly increased parameter counts, so if we want improved performance at matched FLOPs it seems like it might be better to use an MoE instead of a three or four stage hierarchy.

---

### Official Review · Reviewer_R5bc · 2025-06-29

**Clarity:** 3
**Significance:** 3
**Originality:** 3
**Rating:** 5
**Confidence:** 5

**Summary:**

Token based LLM's by design add biases towards languages due to the fixed pretrained vocabulary.
The paper follows a series of byte/ fixed-word-split processing and lifts findings to a multi-stage hierarchical solution, where the most-inner block, on which most parameters and compute are spent, aggregates even several words.
As such it is a significant improvement towards unbiased (i.p. w.r.t. low-ressource languages) and more efficient LLMs.

**Questions:**

- Q1 please address above's weaknesses
- Q2 i seem to miss sth-  in table 2 you show 51% MMLU on AU-net2 at 1e22, compared to baseline 49.2 at 9e21, but it appears to look worse in figure 3 or am i just underestimating the exp-scale axis??

**Ethical Concerns:**

["NO or VERY MINOR ethics concerns only"]

**Final Justification:**

the research is a solid addition to the 'subword-trained tokenizer free LLM's' research domain.
it is not groundbreaking or overly novel but still important and potentially impactful shaping the future of more efficient LLM architectures.
as such i keep my initial solid accept rating

**Limitations:**

as written in W3, please add code to limitations -
other fixed word-split rules showed degradation in math and code, for which it might be hard to find adequate regex-split rules (however you might outperform current SOTA with your 4-word aggregation? unfortunate that you couldn't show that.)

**Quality:**

3

**Strengths And Weaknesses:**

- S1 very well written, visualized, easy to follow
- S2 thorough experiment series, claim of some stable scaling law hyperparameters (albeit i'm missing appendix w/ extensive discussion)
- S3 code is gonna be made public
- S4 highly relevant field towards efficient/ unbiased LLMs


- W1  a/ Results of Table 1 should be discussed somewhere
       b/ I don't seem to find descriptions of what AU-Net 1/2/3/4 are precisely (with dimensions)
- W2 a/ you claim to find stable optimization hyperparameters (line 158ff) but i can only see some theoretic deviation - i expect further discussion in appendix with sweep results.
     b/ i might have missed it but you do not seem to argue how you derive to your embedding dimensions - how/why did you choose 512/ 2048/ 3072 in Fig1 and 3 layers for the outer ones? any discussion of evidence on optimality from sweeps etc highly appreciated. Is this not required for a/ as well to be able to argue abt stable parameters?
- W3 please add "code" to limitations/discussion
- W4 why do you not show AU-net 3/4 for 8b in table 2?

---

> ### Author Rebuttal · Authors · 2025-07-30
>
> We thank the reviewer for this  careful and constructive review, and answer below the questions raised.
> ---
> > **a/ Results of Table 1 should be discussed somewhere**
> > **b/ I don't seem to find descriptions of what AU-Net 1/2/3/4 are precisely (with dimensions)**
>
> Table 1 was intended as a quick summary. We’ll add a short discussion in the introduction to contextualize the numbers and summarize key take-aways. AU-Net 1 was a typo and has been updated to AU-Net 2.
> AU‑Net 2/3/4. Architectural details are delegated to the Appendix due to lack of space, but we will add some basic details (such as number of layers and embedding dimensions) to the main text.
>
> > **a/ you claim to find stable optimization hyperparameters (line 158ff) but i can only see some theoretic deviation - i expect further discussion in appendix with sweep results.**
>
> Thank you for pointing this out—we will add the hyperparameter sweep results and further discussion in the Appendix.
>
> > **b/ i might have missed it but you do not seem to argue how you derive to your embedding dimensions - how/why did you choose 512/ 2048/ 3072 in Fig1 and 3 layers for the outer ones? any discussion of evidence on optimality from sweeps etc highly appreciated. Is this not required for a/ as well to be able to argue abt stable parameters?**
>
> Our first ablations revealed that the first level does not need to be large, while later levels benefit from more capacity, so that more compute is spent on increasingly abstract embeddings. This informed choices like 512/2048/3072 and three outer layers (see Appendix C.2). We will explain further the rationale behind first stage embedding size in the Appendix.
>
> > **please add "code" to limitations/discussion**
>
> We will add an explicit limitation regarding code/math. We focused on DCLM to enable clean comparisons. Training this architecture on more diverse code and math data to evaluate reasoning and generation‑heavy workloads is an exciting direction for future work.
>
> > **why do you not show AU-net 3/4 for 8b in table 2?**
>
> Due to our limited compute budget.. We agree these experiments are valuable and will add them when resources permit.
>
> > **please address above's weaknesses**
>
> Covered above: we will (i) discuss Table 1 in the intro, (ii) add architectural details and dimensions for AU‑Net 1–4, (iii) include sweep results and discussion of hyperparameter stability and design choices in the appendix, (iv) add the code/math limitation, and (v) clarify the compute constraint behind missing AU‑Net 3/4 at 8B.
>
> > **i seem to miss sth- in table 2 you show 51% MMLU on AU-net2 at 1e22, compared to baseline 49.2 at 9e21, but it appears to look worse in figure 3 or am i just underestimating the exp-scale axis??**
>
> The models at 7B in Table 2 operate in a different data-to-model ratio (5 in Table 2 and 10 in scaling laws), and the scaling laws had the most recent hyperparameter sweep values, resulting in better performance for the baseline and slightly better for AU-Net. MMLU is especially sensitive to learning rate. (All the detailed hyperparameters are in Appendix D.) We will adjust the table captions accordingly.

---

> > ### Comment · Reviewer_R5bc · 2025-08-03
> >
> > i thank the authors for the response and am looking forward to the revised version.
> > i keep my good scores as i still see them fit.

---

### Official Review · Reviewer_Wegy · 2025-06-30

**Clarity:** 3
**Significance:** 3
**Originality:** 3
**Rating:** 5
**Confidence:** 4

**Summary:**

The paper describes a novel U-Net-inspired language model architecture operating directly on raw byte representation, without requiring a predefined (e.g., BPE) tokenizer.  Instead, the paper proposes a hierarchical architecture where components of the network operate on different temporal scales (by pooling subsequences of the input byte-level tokens), in the hope of more efficiently capturing
longer context and operating at higher level understandings.

In contrast to similar previous work like Hierarchical Transformer or MegaByte, the pooling hierarchy is not based on fixed window size, but is instead derived from the underlying content, e.g., splitting on words and ngrams.  Experiments show that the proposed model can achieve comparable or improved performance a vanilla BPE-token Transformer baseline, although performance improvements on some tasks (e.g., MMLU) require more training compute than the BPE baseline.

**Questions:**

Specific questions:

C.f. Sec 4, the key innovations compared to previous work on byte-level LLMS, appear to be 1) the more semantically motivated splitting function, and 2) training at larger scale compared to previous work (e.g., [4]) which seems to be important for byte-based models (cf. Fig. 3).

I think the paper would be *greatly* strengthened if it more explicitly addressed 1 by including explicit comparisons of the proposed word-based splitting to simpler baselines
  (e.g., fixed window size at each level, as previously done in the literature), keeping the architecture, scale, and training data constant.
  Acknowledging of course that this might be difficult to achieve in limited time, such a comparison to quantify the importance of the proposed splitting function would better highlight the novel aspects of the paper and and strengthen the its contribution.

Similarly, although less critical, an assessment of a byte level transformer baseline (i.e., no hierarchy) missing from the key results in table 1.

Minor issues:

- Contribution C2 is a bit overstated.  This advantage should apply to any "byte-level" language model, and is not unique to this work.  E.g., Byt5 (https://arxiv.org/abs/2105.13626) or cited baselines such as MegaByte [17], BLT [5], etc.

- I'm confused by the $\gamma_{byte}$ equation after line 147.  Why is there a fixed scaling factor (1/k) between F_{model/byte} and F_{model/token}?  Shouldn't this be a function of the actual architecture being used (as in the following paragraph about AU-Net)?

- The text uses somewhat misleading language about the representation at deeper levels of the hierarchy "predicting further ahead", similar to multi-token prediction (Fig. 1 and Sec. 1), which is, to my understanding, imprecise.
  It seems true that the pooled representations in deeper stages can more efficiently take longer context into account -- they essentially effectively broader "receptive field" per token compared to shallower stages.
  But, assumiung fully causal self attention and pooling/expansion (as show in Fig. 1), a network trained only to do next-byte-token prediction is not actually predicting any further *ahead*.  To the contrary, it is looking further back (per-timestep).

- Table 1 appears to be mislabeled: the AU-Net 1 and 2 rows contain the same results as the rows labeled AU-Net 2 and 3, respectively in Table 2.

- The notation used in the "Hyperparameter scaling laws" section (lines 158-164) is not clearly defined.  What is BSZ (batch size?)?
  - Also, the first mention of DCLM (line 144) should probably include a reference.

**Ethical Concerns:**

["NO or VERY MINOR ethics concerns only"]

**Final Justification:**

In their rebuttal the authors committed to including a fair comparison to a fixed-window baseline in the final camera-ready draft, and responded to my other minor concerns.  In light of this I've increased my rating to 5.

**Limitations:**

Yes

**Paper Formatting Concerns:**

No concerns.

**Quality:**

3

**Strengths And Weaknesses:**

Strengths
- Clearly written
- Novel and well-motivated data-dependent "splitting function" approach
- Strong performance: roughly comparable to traditional BPE baseline when using a single level of splitting (at word boundaries), improving with increased levels of hierarchy
- Comprehensive evaluation, including scaling analysis

Weaknesses
- Experiments feel a bit incomplete: there are no explicit comparisons between different splitting functions, e.g., to a fixed-window-size baseline.  Given that one of the primary contributions is the use of content-adaptive splitting (as byte-level LLMs have been trained before), it would be informative to quantify the importance of the choice.
- The proposed splitting approach is specific to languages which are easily split into words, it is unclear how to generalize well to broader language where this is more difficult such as Chinese or Japanese.

---

> ### Author Rebuttal · Authors · 2025-07-30
>
> We thank the reviewer for this  careful and constructive review, and answer below the questions raised.
> ---
>
> > **I think the paper would be greatly strengthened if it more explicitly addressed 1 by including explicit comparisons of the proposed word-based splitting to simpler baselines (e.g., fixed window size at each level, as previously done in the literature), keeping the architecture, scale, and training data constant.**
>
> We agree this is important. We had early runs that favored dynamic, content‑adaptive pooling, so we allocated more compute there. If the paper is accepted, we commit  to include a fixed‑window comparison under the same architecture/scale/data by camera‑ready, as running these experiments will take time.
>
> > **The proposed splitting approach is specific to languages which are easily split into words, it is unclear how to generalize well to broader language where this is more difficult such as Chinese or Japanese.**
>
> We agree. Our current scope focuses on languages with a clear word segmentation. As acknowledged in the limitations section, extending our method to scripts without explicit word boundaries (e.g., Chinese, Japanese, Thai) remains an open question.
>
> > **Similarly, although less critical, an assessment of a byte level transformer baseline (i.e., no hierarchy) missing from the key results in table 1.**
> Table 2 has a byte level (no hierarchy) comparison for a smaller amount of data (80B tokens). Running a byte level model on 370B llama3 tokens would be more than 4 times as expensive than AU-Net or BPE model, which would be too much compute for a model of this size.
> > **Contribution C2 is a bit overstated. This advantage should apply to any "byte-level" language model, and is not unique to this work. E.g., Byt5 (https://arxiv.org/abs/2105.13626) or cited baselines such as MegaByte [17], BLT [5], etc.**
>
> Agreed—this advantage is common to byte‑level models (e.g., ByT5, MegaByte, BLT). By showing parity with a strong BPE baseline with minimal trade‑offs, we provide concrete evidence of its practical value. We will soften C2 accordingly.
> >  **equation after line 147. Why is there a fixed scaling factor (1/k) between F_{model/byte} and F_{model/token}? Shouldn't this be a function of the actual architecture being used (as in the following paragraph about AU-Net)?**
>
> The equations map token‑level to byte‑level scaling for any byte‑level LM. They are architecture‑agnostic: each model can have a different FLOPs/byte, but the conversion itself doesn’t depend on the specific architecture. We will clarify this point.
> > **The text uses somewhat misleading language about the representation at deeper levels of the hierarchy "predicting further ahead", similar to multi-token prediction (Fig. 1 and Sec. 1), which is, to my understanding, imprecise. It seems true that the pooled representations in deeper stages can more efficiently take longer context into account -- they essentially effectively broaden "receptive field" per token compared to shallower stages. But, assuming fully causal self attention and pooling/expansion (as shown in Fig. 1), a network trained only to do next-byte-token prediction is not actually predicting any further ahead. To the contrary, it is looking further back (per-timestep).**
>
> You are correct that when taken as a whole, our model only does next-byte prediction. However, outputs of deeper stages influence the prediction of multiple bytes in the future. All layers are causal but at different levels of granularity. Stage 1 outputs predictions that directly predict the next byte, Stage 2 outputs vectors that help the prediction of the next word, Stage 3 outputs vectors that help the prediction of the next two words. In this sense, Stages 2 and 3 have to predict further in the future. Finally, stages 2 and 3 represent the vast majority of the total compute of the model, which forces it to rely less on short term byte level predictions and more on word and word group level predictions.  We will clarify this section to make it more precise.
> > **Table 1 appears to be mislabeled: the AU-Net 1 and 2 rows contain the same results as the rows labeled AU-Net 2 and 3, respectively in Table 2.**
>
> Thank you, we will fix this.
> > **The notation used in the "Hyperparameter scaling laws" section (lines 158-164) is not clearly defined. What is BSZ (batch size?)?**
>
> We will define BSZ (“batch size”) on first use.
> > **Also, the first mention of DCLM (line 144) should probably include a reference.**
>
> We will cite DCLM at its first mention.

---

> > ### Comment · Reviewer_Wegy · 2025-08-03
> >
> > I appreciate the author's response to my comments.
> >
> > > If the paper is accepted, we commit to include a fixed‑window comparison under the same architecture/scale/data by camera‑ready, as running these experiments will take time.
> >
> > I look forward to seeing this.
> >
> > >> equation after line 147. Why is there a fixed scaling factor (1/k) between F_{model/byte} and F_{model/token}? Shouldn't this be a function of the actual architecture being used (as in the following paragraph about AU-Net)?
> > > The equations map token‑level to byte‑level scaling for any byte‑level LM. They are architecture‑agnostic: each model can have a different FLOPs/byte, but the conversion itself doesn’t depend on the specific architecture. We will clarify this point.
> >
> > I guess I must be missing something.  Upon reflection, I think I was just confused by the decomposition of $C$.
> >
> > It makes sense to me that the compute ratio between a token and byte decoder-only transformer will scale with $k^2$ -- due to the quadratic self attention operation on sequences whose lengths scale with a factor of $k$.
> >
> > But (at least in my reading) the text seemed to imply that $F_{model/input-unit}$ is independent of the sequence length, which it is not for self-attention models (so it does actually depend on the architecture...).  I recommend editing this section to explain the reason why $F_{model}$ differs between bytes and tokens more explicit.
> >
> > > You are correct that when taken as a whole, our model only does next-byte prediction. However, outputs of deeper stages influence the prediction of multiple bytes in the future. All layers are causal but at different levels of granularity. Stage 1 outputs predictions that directly predict the next byte, Stage 2 outputs vectors that help the prediction of the next word, Stage 3 outputs vectors that help the prediction of the next two words. In this sense, Stages 2 and 3 have to predict further in the future.
> >
> > We're in agreement about the architecture.  But I think my point still stands, if perhaps a bit nitpicky.  The activations of a transformer layer *all* contribute to future predictions (via self attention in the subsequent layer).  But that doesn't make them 'predictions' themselves -- we don't really know how to interpret activations within these complicated neural networks.  In particular I found the analogy with multi-token prediction to be a bit of a stretch.  My recommendation was simply to soften the language to describe granularity of the representation, and the (useful!) inductive bias it implies.
> >
> > > Finally, stages 2 and 3 represent the vast majority of the total compute of the model, which forces it to rely less on short term byte level predictions and more on word and word group level predictions. We will clarify this section to make it more precise.
> >
> > Indeed this seems like a valuable point to highlight!

---

> > > ### Author Response · Authors · 2025-08-05
> > >
> > > Thank you for taking the time to answer our rebuttal and pointing out sections that are unclear or imprecise. We will rephrase and clarify the scaling section’s formulas to frame it purely as a conversion of units.
> > >
> > > We agree that saying that the model predicts multiple tokens in the future is speculative at best. Deeper stages predict vectors that ultimately end up as bytes of the next word. We will highlight that most of the compute is being spent at the word or word group level.
> > >
> > > Thank you again for the constructive feedback!

---

### Official Review · Reviewer_SrNB · 2025-07-02

**Clarity:** 2
**Significance:** 3
**Originality:** 4
**Rating:** 5
**Confidence:** 3

**Summary:**

The paper introduces autoregressive U-Net (AU-Net), an architecture for language modeling that substitutes the standard tokenization and embedding steps with a learnable hierarchical embedding process applied directly on raw bytes. AU-Net has contracting and expanding steps. Stage 1 processes raw bytes, Stage 2 pools at word boundaries, Stage 3 at two-word chunks, and Stage 4 at four-word chunks, with deeper stages naturally predicting further into the future and capturing broader semantic patterns while earlier stages handle fine-grained details. Skip connections allows access to information from previous stages. The expanding path predicts text sequences by transforming coarse information back into fine-grained representations. The authors show that within controlled compute budgets, AU-Net matches or exceeds BPE-based Transformer baselines on standard benchmarks, with deeper hierarchies showing promising scaling trends and particular advantages on character-level tasks and multilingual benchmarks, especially for low-resource languages that benefit from byte-level processing.

**Questions:**

- Why have you chosen an english-centric dataset (DCLM) to train the model?

- Stage 1 processes raw bytes via a self-attention mechanism, that creates byte-level embeddings before Stage 2 pools at word boundaries. How exactly are these raw bytes fed into the self attention mechanism?

Typos:

l59: missing a period "." after "Practical Efficiency"

l118: missing a space between "two words" and "(or sentence..."

**Ethical Concerns:**

["NO or VERY MINOR ethics concerns only"]

**Final Justification:**

My minor weaknesses have been mainly addressed in the rebuttal. I believe the paper presents a relevant contribution that is worth publishing.

**Limitations:**

yes

**Quality:**

3

**Strengths And Weaknesses:**

Strengths

- The paper uses careful compute-controlled comparisons enabling fair comparisons across architectures.

- AU-Net achieves strong results when compared with BPE-based Transformer baselines, with scaling law results demonstrating that 2-stage and 3-stage variants consistently match BPE baseline performance across different compute budgets and benchmark tasks.

- The paper shows strong results in multilingual settings despite using an English-centric dataset (DCLM), with consistent improvements on low-resource language translation tasks, demonstrating the advantages of byte-level processing for cross-lingual transfer.

Weaknesses

- The use of an English-centric dataset (DCLM) may unfairly disadvantage BPE baselines, which are specifically designed to handle diverse vocabularies. The authors don't justify why they chose DCLM over more multilingual datasets, making it unclear how AU-Net would compare with Transformer + BPE baselines on multlingual benchmarks given standard multilingual training corpus.

- The paper lacks precise mathematical formulations of the model architecture, relying primarily on prose descriptions. Stating the actual computations performed at different stages and modules would help to better understand the architecture.

- Not a big weakness, but many languages in the FLORES-200 evaluation are closely related variants within the same language families (e.g., Spanish, Asturian, Catalan, Galician for Romance; or various Italian dialects), which might inflate the apparent multilingual improvements.

- Although acknowledged by the authors, the lack of support for non-space-delimited languages represents a fundamental architectural limitation.

---

> ### Author Rebuttal · Authors · 2025-07-30
>
> We thank the reviewer for this thorough review, and answer below the questions raised.
>
> ---
>
> > **The use of an English-centric dataset (DCLM) may unfairly disadvantage BPE baselines, which are specifically designed to handle diverse vocabularies. The authors don't justify why they chose DCLM over more multilingual datasets, making it unclear how AU-Net would compare with Transformer + BPE baselines on multilingual benchmarks given a standard multilingual training corpus.**
>
> We chose DCLM to ensure comparability with existing literature and because it was the state-of-the-art pretraining dataset when starting this work. While BPE can be trained multilingually, most popular tokenizers are English-centric, and BPE's pre-tokenization rule often splits on spaces, similar to our approach. Our model's ability to use learnable embeddings instead of a fixed embedding table offers greater flexibility, essentially providing an infinite vocabulary. We agree that exploring multilingual and code datasets in future work would be highly valuable.
>
> > **The paper lacks precise mathematical formulations of the model architecture, relying primarily on prose descriptions. Stating the actual computations performed at different stages and modules would help to better understand the architecture.**
>
> We apologize for the lack of precise mathematical formulations in the paper. We will add formulas for the downsampling and upsampling operations for clarity and completeness, while still referring to the code for more detailed implementation.
>
>
> > **Not a big weakness, but many languages in the FLORES-200 evaluation are closely related variants within the same language families (e.g., Spanish, Asturian, Catalan, Galician for Romance; or various Italian dialects), which might inflate the apparent multilingual improvements.**
>
> We acknowledge this point regarding the relatedness of some languages in the FLORES-200 evaluation. Our aim was to demonstrate the effectiveness of byte-level processing for cross-lingual transfer, especially for low-resource languages, and the results still indicate a consistent improvement over baselines, even if some languages are closely related.
>
> > **Although acknowledged by the authors, the lack of support for non-space-delimited languages represents a fundamental architectural limitation.**
>
> We agree that supporting non-space-delimited languages (e.g., Chinese, Japanese, or Korean scripts) remains a limitation. We have designed the splitting implementation to be quite generic, allowing future work to define new splitting functions tailored to these languages. We have recently discovered the existence of the uniseg python package that uses dictionaries to segment text into words in many languages without relying purely on spaces.
>
> > **Stage 1 processes raw bytes via a self-attention mechanism that creates byte-level embeddings before Stage 2 pools at word boundaries. How exactly are these raw bytes fed into the self-attention mechanism?**
>
> Raw bytes are obtained by decoding a string with UTF-8, which produces a sequence of bytes. These bytes are then fed into the self-attention mechanism after being embedded using a 258-entry embedding table, which includes Beginning-of-Sentence (BOS) and End-of-Sentence (EOS) as special characters.

---

> > ### Comment · Reviewer_SrNB · 2025-08-06
> > **Response to Rebuttal**
> >
> > Thank you for addressing my concerns. I believe the paper presents a relevant contribution that is worth publishing, and that my rating reflects this.

---

### Decision · Program_Chairs · 2025-09-17

**Decision:**

Accept (poster)

**Comment:**

***Summary***

The authors introduce a hierarchical transformer called AU-Net, with multiple layers of hierarchy.  Stage 1 operates on bytes/characters.  Characters are pooled into words, using an external segmenter, which segments on spaces.  In contrast to some previous published work, which uses fixed-stride patches, AU-Net allows words to vary in length.  Stage 2 then operates on words; it is a larger transformer model, with a larger embedding dimension and greater parameter count.  Stages 3 and 4 operate similarly, but are fixed-stride; they pool words into pairs of words or quad-word chunks, since no segmenter is available to find meaningful phrase boundaries.  After the encoding/pooling stages, there are a series of decoding/unpooling stages, modelled after the U-Net architecture, with skip connections between encoder and decoder.

The authors compare their architecture against a standard transformer using byte-pair-encoding (BPE), and are able to match or slightly exceed the BPE baseline.

***Meta-review***

The reviewers were split, with ratings (2, 5, 5, 5).  Although most reviewers voted to accept, reviewer T1Se (rating 2) actually had a detailed and insightful review in my opinion.  I personally would give this paper a rating of 4 -- "borderline accept".

Selected comments from the reviewers can be found below.  Most reviewers were impressed by the experimental results, as was I.  However, there was broad agreement that the use of a word-splitting function based on whitespace was a major limitation, since it does not work for languages like Chinese or Japanese.  There was also broad agreement that there should be comparisons against additional baselines, particularly against other hierarchical transformers, of which several have been published in the literature.  I agree wholeheartedly with both of these criticisms.

***AC opinion***

This is an area of research that I am actually quite familiar with, so I would like to make the following additional observations.  On the positive side, this paper is well-written, and the experiments were carefully done.  The experimental results are decent, matching the BPE baseline even when controlling for parameter count, which I have not seen before.  I recommend "accept" mainly on the strength of those results.

Moreover, the limitations of BPE tokenizers are also well known, and are a major source of frustration across the industry, so a truly good "neural tokenizer" would be an extremely high-impact paper, worthy of a spotlight or oral.  Unfortunately, this is not such a paper, for several reasons.

First, the novelty of the author's architecture is very limited.  Hierarchical transformers have been proposed before.  Transformers that segment on variable word lengths have been proposed before (e.g. Neitemeier et. al. ICLR 2025).  UNet-style architectures have been proposed before.  This paper has, at best, minor difference in pooling/unpooling from prior published work.  The author's "related work" section is extremely minimal; it cites the necessary literature, but does not provide much discussion.  Much like the reviewers, I would really like to see a comparison against this prior work.

Second, there are hardly any ablations.  A paper which did a proper survey of the design space, comparing fixed-stride/variable-stride, different segmenting strategies, and different pooling/unpooling strategies would be much more impressive.  Unfortunately, the authors do no such thing.  They do not even compare their whitespace segmentation against an entropy-based or BPE segmentation, which previous publications have done.  Moreover, as the reviewers mention, whitespace segmentation only works on European languages, and thus has limited applicability.

Third, the major implementation issue with variable-stride segmentation is that batched inference becomes problematic, because the word boundaries do not align between different sequences in the batch.  The authors do not even discuss this point as a limitation.  Unfortunately, none of the reviewers mentioned this either, but it would be a major obstacle to deploying AU-Net in the real world.

Fourth, while it's great that the authors test on downstream tasks, I am not fully convinced.  I would actually like to see perplexity numbers as well, converted to bit-per-character, since that is where I've seen previous efforts at tokenizer-free models fall short.

***Selected comments from reviewers.***

Reviewer SrNB (rating 5):
* "The paper shows strong results in multilingual settings despite using an English-centric dataset (DCLM)"
* "Although acknowledged by the authors, the lack of support for non-space-delimited languages represents a fundamental architectural limitation."

Reviewer Wegy (rating 5):
* "Strong performance: roughly comparable to traditional BPE baseline when using a single level of splitting"
* "Experiments feel a bit incomplete: there are no explicit comparisons between different splitting functions, e.g., to a fixed-window-size baseline."
* "The proposed splitting approach is specific to languages which are easily split into words, it is unclear how to generalize well to broader language where this is more difficult such as Chinese or Japanese."

Reviewer R5bc (rating 5):
* "the research is a solid addition to the 'subword-trained tokenizer free LLM's' research domain. it is not groundbreaking or overly novel but still important and potentially impactful"

Reviewer T1Se (rating 2):
* "The paper is mostly well written and easy to follow."
* "It is not clear how AU-Net-2 differs from prior work using adaptive pooling heuristics, e.g. the whitespace-split setting of BLT [3] and DTP [4] (DTP should also be referenced)."
* "From the paper, it seems like AU-Nets vastly outperform BLT with similar FLOPs/token budgets. My question is: where is this coming from?"
* "Although scaling the levels of the hierarchy (AU-Net-3 and AU-Net-4) slightly improves performance in a FLOP-matched training setup, it seems like a problematic dimension to scale due to the vastly increased parameter count...  if we want improved performance at matched FLOPs it seems like a better choice to use a MoE instead"
* "(1) we already know from prior work (e.g. [1]) that a two-stage architecture can be beneficial and (2) the authors' results for going beyond two stages seem discouraging to me"